# Equity of Social Health Insurance Coverage for Migrants in Thailand: A Concentration Index Analysis

**DOI:** 10.3390/ijerph19010064

**Published:** 2021-12-22

**Authors:** Peeraya Piancharoen, Hathairat Kosiyaporn, Rapeepong Suphanchaimat

**Affiliations:** 1International Health Policy Program (IHPP), Ministry of Public Health, Nonthaburi 11000, Thailand; hathairat@ihpp.thaigov.net (H.K.); rapeepong@ihpp.thaigov.net (R.S.); 2Division of Epidemiology, Department of Disease Control, The Ministry of Public Health, Nonthaburi 11000, Thailand

**Keywords:** equity, migrant workers, income-related inequality, inequality decomposition, social security scheme, concentration index, concentration curve, Thailand

## Abstract

Thailand is attempting to implement an effective health insurance scheme to cover all migrant workers in the country. One of the remarkable policies is the Social Security Scheme (SSS). This study aims to assess the equity of SSS coverage among migrant workers in Thailand, sorted by types of businesses (agriculture, services and industrial sectors) and Gross Provincial Product (GPP) per capita. A secondary data analysis on time series cross-sectional data was employed. The dataset comprised: (1) the number of migrant insurees under the SSS; (2) the volume of migrant insurees in formal and informal sectors; and (3) provincial economic level and provincial population by years from 2015–2018. Descriptive statistics, Spearman’s rank correlation, and concentration index analysis on the ratio of SSS insurees to all migrants ranked by GPP per capita and business types from 2015–2018 were performed. Results showed that the ratio of SSS insurees to all migrants increased from 38.5% to 58.9%. Spearman’s correlation found a positive relationship between the SSS coverage and GPP per capita and business types in 2018. The statistical significance (*p* < 0.001) was found only in the industrial sector (r_s_ = 0.346). Significant CIs were found in SSS coverage in the industrial sector in 2016 (CI = 0.147, *p* < 0.001), and SSS coverage in the industrial sector in 2017 (CI = 0.137, *p* < 0.001). In conclusion, the trend of CIs displayed a movement towards zero for all business types. This implied that the distribution of migrants in SSS shifted toward the equitable distribution across provinces in Thailand. A possible explanation was a major change in Thai politics in 2014 and a change in employment legislation for migrants in 2017. To expand the insurance coverage, the government should use the CI as a guide to consider specific provinces or target specific economic sectors as a priority to expedite the insurance enrolment for migrants.

## 1. Introduction

Migrant health has been recognised as a global concern over decades [1]. According to the International Organization for Migration (IOM), Thailand is one of the major destination countries for international migrant workers in Southeast Asia (SEA) with the total number of migrants in the country numbering about 3.6 million in 2019 [2]. They are mostly from Myanmar, Lao PDR, Cambodia, and Vietnam (so-called CLMV nations), working in the construction, agriculture, and fishery sectors. The pool of migrant workers accounted for 10% of the entire national labour force [3].

To protect the health of migrants in Thailand, one of most prominent policies of the Thai government is the promulgation of national insurance scheme for migrants. There are two main schemes. The first is the Social Security Scheme (SSS). The SSS is managed by the Social Security Scheme of the Ministry of Labour. It is a compulsory scheme financed by tripartite contributions: government, employers, and migrant workers. All migrant workers in the formal sector (like factory workers or workers in business enterprises) are obliged to be covered by the SSS. Enrolment in the SSS is indicated in the employment law (Social Security Scheme Act B.E.2533) [4]. The second scheme is the Health Insurance Card Scheme (HICS), a semi-compulsory scheme managed by the Ministry of Public Health (MOPH) [5]. The MOPH encourages migrant workers who are not covered by the SSS (for instance, seasonal agricultural and fishery workers, domestic workers, and hawkers) to purchase the insurance under the HICS. However, there is no penalty on migrant workers who refuse to purchase the HICS, as well as the employers of these migrants (making the nature of the scheme semi-compulsory) [3].

A remarkable change in the law related to migrant health is found in 2014 after the coup d’état [6]. There was a high outflow of migrant workers returning to their home countries leading to the deficit of labour force Thailand. The One Stop Service (OSS) measure was introduced to resolve the exodus of migrants by allowing migrants to register with the government and proceed to national verification for the undocumented migrants [7]. Those registered could then obtain a work permit and become eligible to enroll in either the SSS or the HICS, based on types of work. Another remarkable change took place again in 2017 when the Royal Ordinance on the Management of Foreign Workers Employment (B.E. 2560) was introduced. The new law imposes a more severe penalty on employers who leave their migrant employees uninsured [8]. Following this law, the number of SSS beneficiaries among migrant workers increased substantially from 357,643 in 2013 to 1,107,426 in 2018 [3]. Despite the existence of legal measures to enroll migrant workers in the SSS, uninsured migrant workers remain. Reasons for this include delayed registration process, avoidance of payroll contributions from employers and migrants, and migrants’ ignorance of the scheme’s existence [9].

Enrollment in the insurance is associated with not only the individual behaviour of migrants, but also the economic structure of the areas where migrants are working. The Migrant Integration Policy Index (MIPEX) 2020 reported that wealthier countries, measured by gross domestic product (GDP) per capita, reported better migrant healthcare coverage and ability to access services [10]. For micro-economic status, there were some previous studies in analysis of association between individual socioeconomic status to healthcare access and health outcomes [11]. It found that migrants with low incomes were most likely to have poor health access and health status. Though there are a few studies that explore the relationship between individual economic status and migrant access to care, to our knowledge, little is known about the association between the macro-economy and insurance coverage for migrants. We hope that this study will provide clearer insights on factors associated with the enrolment of migrants in the insurance scheme and will serve as important inputs to monitor if the insurance policy equitably functions in areas with different economic levels. Therefore, this study aims to assess the equity of SSS coverage among migrant workers in Thailand, sorted by types of businesses (agriculture, service, and industrial sectors). Note that for this study, we focused on the SSS only as it is a compulsory scheme that has a direct obligation on migrant workers and employers whereby for the HICS, as mentioned earlier, the nature of the scheme is still semi-compulsory.

## 2. Materials and Methods

### 2.1. Data Sources and Study Design

This study applied a secondary data analysis on time series cross sectional data. The unit of analysis was province. Note that Thailand is divided into 77 provinces. Thailand’s administrative areas start from subdistrict as the smallest unit, then to district and province respectively. The dataset was composed of (1) the number of migrant insurees under the SSS in Thailand; (2) the volume of migrants in both the formal and the informal sectors; and (3) data on provincial economic level and provincial population by years. We obtained the first dataset (number of SSS insurees) from the Social Security Office and the second dataset (number of migrants) from the Foreign Workers Administration Office, Ministry of Labour (MOL). The provincial-economic and population data were acquired from the National Economic and Social Development Council. We focused on the data only between 2015 and 2018 as the data before and after this period were not complete enough to perform the analysis and focused on SSS coverage only as the SSS is a compulsory scheme, unlike the HICS.

### 2.2. Data Management and Analysis

The analysis was composed of four parts. First, for the SSS data, as the data were arranged in a monthly format while the economic data were stored in a yearly format, we used the median number across months within a year to represent the number of migrants in a respective year. Then we merged the economic data with the yearly migrant data. The study years lasted from 2015 to 2018. The provincial economic data consisted of gross provincial product (GPP) per capita and the GPP per capita sorted by business sectors, namely, agricultural product per capita, industrial product per capita, and service product per capita.

Second, we conducted descriptive statistics to present the overview of number of migrants in Thailand, number of SSS migrants and the ratio of SSS insurees to all migrants (we named the ratio as SSS coverage). Additionally, we illustrated the descriptive statistics of economic data across provinces in Thailand.

Third, we performed the association between SSS coverage and each provincial economic level during 2015–2018 by Spearman’s rank correlation [12].

Fourth, we assessed the equity of insurance coverage respective to the provincial economic level by concentration index (CI). The CI is widely recognized as a useful tool to evaluate the degree of socioeconomic related inequality in health. The CI was acquired by regressing 2α^2^_r_ (h_i_/μ) on r_i_, as proposed by the World Bank [13,14] where the coefficient β represented the point estimate of the CI, α^2^_r_ denoted the variance of the fractional economic rank, h_i_ was the health variable of interest (in this case, insurance coverage), μ was the mean of h_i_, and r_i_ was the rank of population unit respective to provincial economic status. A statistical significance was determined at the 95% confidence level. We used Stata software v16.1 (serial license number: 301606241761) for all calculations. The positive value (from 0 to +1) of CI meant the insurance coverage was concentrated amongst the better-off provinces. On the other hand, there would be a negative value (from 0 to −1) if the insurance coverage was concentrated amongst the less economically well-off provinces. The results were also graphically displayed by concentration curves (CC). The CC would lie above the 45-degree equality line if the CI took a negative value and would lie below the equality line if otherwise.

## 3. Results

### 3.1. Numbers of Migrants and SSS Insurees

Overall, the average number of migrants in each province in Thailand increased from 19,527 to 29,224 during 2015–2017. The median number of migrants in 2018 was 71,114, about double the figure in 2015. Note that the range of the number of migrants was extremely wide. For instance, in 2018, the minimum and maximum numbers of migrants in a province were 216 and 459,118 respectively. The distribution of SSS insurees and migrants varied by provinces. The number of SSS insurees increased dramatically in most provinces in 2018 (Figure 1), a year after the promulgation of the new employment law in 2017, which raised the penalty for employers who left their migrant employees unregistered. The total volume of migrant workers from a macro-perspective remained stable (Figure 2).

The ratio of SSS insurees to all migrants also increased markedly from 38.5% in 2015 to 58.9% in 2018. The increasing trend was also observed in the volume of SSS insurees. Standard deviation of ratio of SSS insurees to all migrants across provinces in 2018 was also far greater than that of other years. The observation that the mean was far larger than the median could infer that the data on the number of migrants and the number of SSS insurees were rightly skewed and thus median was a more appropriate measure of frequency (Table 1).

### 3.2. Economics Variables

Focusing on economic status, the mean provincial GPP per capita increased from 146,575.75 (US$4328.5) Baht in 2015 to 169,506.70 Baht (US$5055.7) in 2018 (Figure 3). The service sector contributed most to provincial GPP per capita (46.96% in 2015 to 48.72% in 2018). The median of the contribution of provincial GPP per capita of the industrial sector was lower than that of the agricultural sector although the mean of the industrial sector was larger.

### 3.3. Spearman’s Correlation

Spearman’s correlation coefficients (r_s_) represented that degree of correlation between the rank of SSS coverage and the rank of provincial GPP. The analysis found positive relationships in all study years for the industrial sector with a statistical significance observed during 2016–2018 (Table 2). The positive relationship meant that the SSS coverage and the economic contribution of industrial business grew in the same direction which could imply that the SSS coverage was greater in areas where industrial business grew. Year 2018 saw a positive value of the coefficients in all business sectors (for instance, the moderate degree correlation (r_s_ = 0.346) of industrial per capita in year 2018 with statistical significance (*p* < 0.001)).

### 3.4. Concentration Curves

The upper left graph of Figure 4 demonstrated the CC of SSS insurance coverage distribution with the rank of provincial economic level. Data were classified by year from 2015 to 2018 compared with a perfect equality line (black line). We observed a different pattern of the CC between year 2015 and years 2016–2018. In 2015, the CC lay above the equality line which represented a relative concentration of SSS insurance beneficiaries in the less well-off provinces. However, the curves in 2016–2017 concavely deviated below the equality line.

The assessment of the SSS coverage against the economic contribution by the industrial, service and agricultural sectors are shown in the other parts of Figure 4. The CC of SSS coverage against the economic contribution by the industrial sector lay convexly above the equality line in 2015. The curves moved closer to the equality line in later years.

In 2015, the CC of SSS coverage plotted against the rank of the provincial economic contribution by the service sector lay furthest above the equality line while the curves in subsequent years skewed below the perfect line. The curve pattern of CC plotted against the economic contribution of the agricultural sector seemed to be different from the curves found in the service and industrial businesses. During 2015–2017, the curve lay above the equality line then moved closer to the equality line in later year. The curve in 2018 dropped to be below the equality line, implying that the insurance coverage was more pronounced in the less well-off provinces.

### 3.5. Concentration Indexes

Figure 5 describes the change of CI ranked by provincial economic status, sorted by different business types (agriculture, industrial and services). The overall trend shifted towards above or close to zero which mean it changed from pro-poor or pro-rich to be more equitable as time passed by. For example, the coefficient in the service sector started from −0.067 in 2015, then enhanced to 0.049 in 2016, and moved towards zero (0.029) or equity in 2018. However, the agricultural sector showed a different pattern of CI compared with other sectors. It commenced from below zero (−0.083) in 2015 then remained steady at about −0.076 to −0.058 between 2016 and 2017. Afterward, it remarkably rose to 0.063 in 2018 which was at nearly the same level as the industrial sector (0.079) and all sectors combined (0.060).

## 4. Discussion

Overall, this study is probably among the first studies in Thailand that examined deeply the relationship between SSS coverage and the provincial economic status. The importance of this research is that it will help the government prioritize areas with certain economic status if policy makers wish to expand the SSS enrolment for migrant workers.

The SSS is part of social protection for workers. This concept is implemented not only in Thailand but also in many countries which rely on a labour-intensive economy. For instance, the Malaysian government introduced the Employment Injury Scheme under the Social Security Organisation (SOCSO) to provide health protection for occupational injuries and diseases, including free treatment at SOCSO panel clinics and government hospitals since January 2019 [15]. For Indonesia, the BPJS Ketenagakerjaan Social Security Scheme was implemented for migrants who worked in Indonesia for at least six months and paid contributions, and it included their family members [16]. Employee registration was required for all social security schemes in the three countries (Indonesia, Malaysia and Thailand); however, penalties for employers who failed to register their employees in the system were clearly written only in Thailand and Indonesia’s Acts. Singapore, another Southeast Asian country, also had an Employment Foreign Manpower Act (EFMA) that regulated employers’ obligations to provide migrant workers with health insurance under social security [16,17]. This study points towards some worthwhile learning and messages. First, though the employment law imposes a penalty on employers who leave their migrant employees uninsured, in some areas the insurance enrollment is still far from complete. Second, the enrollment in the insurance is not subject to the law enforcement alone as the contextual economic environment also matters.

The above findings indicate that the distribution of migrants in the SSS scheme (SSS coverage) was mainly equitable across provinces in Thailand (CIs are close to zero) with some variations from year to year. Some potential explanations of this phenomenon are as follows. In mid-2014, there was a major change in Thai politics. First, the coup took place and at that time there was a major change of the migrant registration policy (namely, the One Stop Service (OSS) policy). The military government launched a leniency period to allow undocumented migrants to register with the government [6,18]. The registered migrants (in the formal sector) were later insured with the SSS as stipulated in the bylaws. As a result, the number of total migrants registered in the system increased exponentially. Note that the OSS policy began in the border areas which were mostly the less well-off provinces. This would explain why the CC lay convexly above the equality line (CI showed negative values) in the early years (2015–2016).

The second explanation was related to the change in the employment law in 2017. In the past, the employment regulation on migrants followed the “Working of Aliens Act, B.E. 2551 (2008) Section 54”, which mentioned that if an employer of migrants did not have his/her employee registered for a work permit, he or she would be fined for 10,000–100,000 baht (US$330–3300) per an unregistered migrant [19]. Then, in 2017, the new law was promulgated. The essence of the law was the same except that the government raised the penalty to 400,000–800,000 Baht (US$11,766–US$23,532) per unregistered migrant. Moreover, the government also increased jail time for undocumented workers and their employers to a maximum of five years [20]. The law was imposed on all types of migrants. In addition, areas that were most affected by the new regulation were likely to be the economically well-off provinces where the provincial economy heavily depended on industrial and service sectors. This might help explain why the CI converged (from negative values) to close to zero in all business sectors in 2018.

Another point worth mentioning is that before 2018 the CI for the industrial sector demonstrated a pro-rich pattern, while the CI for the agricultural sector was reported as pro-poor. This issue draws a point for policy consideration which is that by focusing on the industrial sector, the SSS coverage was concentrated among the rich provinces. In contrast, in the agricultural sector, the SSS coverage was concentrated among the less affluent provinces. The situation also alludes to the fact that the law requiring agricultural migrant workers to be enrolled in the SSS was poorly enforced [21]. This contradicted the primary policy intention that the employment law would be effectively enforced nationwide which is why we initially expected an equitable distribution of the SSS coverage across provincial economy (near-zero CI).

Concerning policy implications, policy makers or public health practitioners may consider using CI or CC to monitor the insurance coverage of migrants in addition to the crude coverage. The analysis may be more granular if sub-provincial data are available. The CI may help prioritize areas for actions and identify which provinces (varying by economic status) are performing well (or poorly) in enrolling migrants to the insurance and which areas need more intensive enforcement of the insurance enrollment than others. Areas where the agricultural business plays a critical role are mostly located in the border areas, while areas close to the country’s centre, including Bangkok and its vicinity, mostly depend on industrial businesses. Future research that explores the geographical effect on insurance coverage may be useful to help tailor policies to promote migrant insurance enrollment.

This article has extended the academic value of research in migrant health. We suggested that the coverage of the insurance is, to some extent, linked with the macro-economic indicators. Most prior research that explored reasons for why migrants dropped out from the public insurance schemes usually pointed to their individual interaction between health providers and migrant themselves. Reasons for this included discrimination, language differences and the lack of support to be integrated in the destination country’s healthcare system.

Despite a comprehensive analysis, some limitations remain. First, the data we retrieved might not be able to reflect “all” migrants in Thailand. Some migrant workers (of unknown number) stay undocumented and untraceable by the government. Most of them are suspected to be in the informal sector rather than the formal sector.

Second, the GPP per capita (though tallied by business sectors) might not serve as a perfect indicator for provincial prosperity. However, it is the only composite indicator we could obtain from the online public domains.

Third, the data of SSS members, the data related to the number of migrants, and the economic data were retrieved from different sources. This would explain why the SSS coverage in certain moments of time exceeded one and this justified why we preferred the term “ratio” over “proportion” when referring to SSS coverage. Such phenomenon might occur because of many reasons. For instance, some migrants’ work permits had already expired, and this excluded them from the SSS, but it was possible that the number of SSS members had not been updated. This issue also pointed towards room for improvement of intersectoral integration of the information system on migrants in the country.

Last is the issue of ecological fallacy [22]. As this research is an ecological study in nature, ecological fallacy is inevitable. That is, the measure of association between variables at the group level might not always represent the association at the individual level. Further studies that explore the relationship between socioeconomic status and insurance status at the individual or household levels are worth exploring. However, this needs huge support from the state authorities and huge effort in data collection processes as a primary survey is required [23].

## 5. Conclusions

From 2015 to 2018, the distribution of migrants in Thailand’s SSS scheme across provincial economic status shifted toward greater equity. SSS coverage distribution for the industrial and services sectors followed a similar trajectory from 2015 to 2018, starting with a pro-rich pattern in 2015 and gradually shifted to an equitable pattern as time passed. In the agricultural sector, however, the distribution moved from pro-poor to equitable within the same period. The equitable distribution could be the result of the major change in Thai politics in 2014 and a change in employment legislation in 2017, both of which had an impact on migrant registration policy and employee behaviour. This study points to the policy implication that despite a legislation to enforce migrant workers to be enrolled in the SSS, the economic contextual environments and location of migrants are also influential factors. Policy makers should use the findings of this study as input to help prioritize areas for scaling up migrant insurance coverage and identifying which economic sectors in different provinces most need an intensive insurance expansion most. Note that there remained some limitations in this study, such as varied data sources and ecological fallacy. Future research is recommended into the sub-level relationship between socioeconomic status and the insurance coverage of migrant workers, as well as research that includes geographic factors in health insurance coverage analysis in Thailand is recommended.

## Figures and Tables

**Figure 1 ijerph-19-00064-f001:**
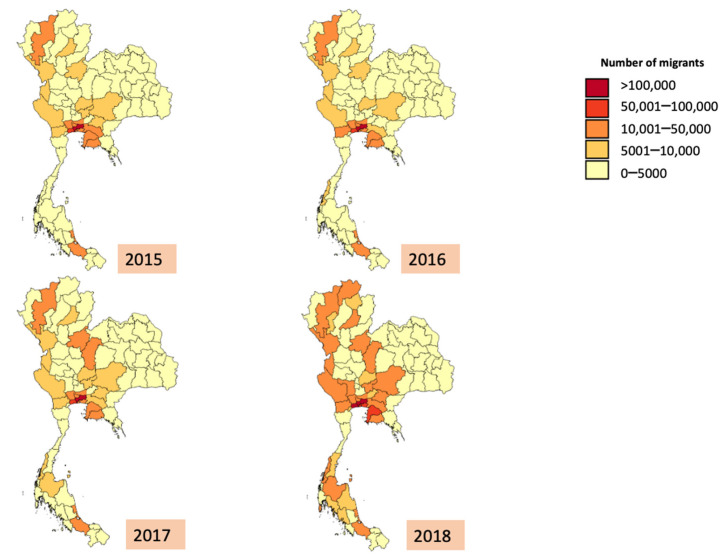
Geographical Difference of migrants’ SSS insurees number, 2015–2018.

**Figure 2 ijerph-19-00064-f002:**
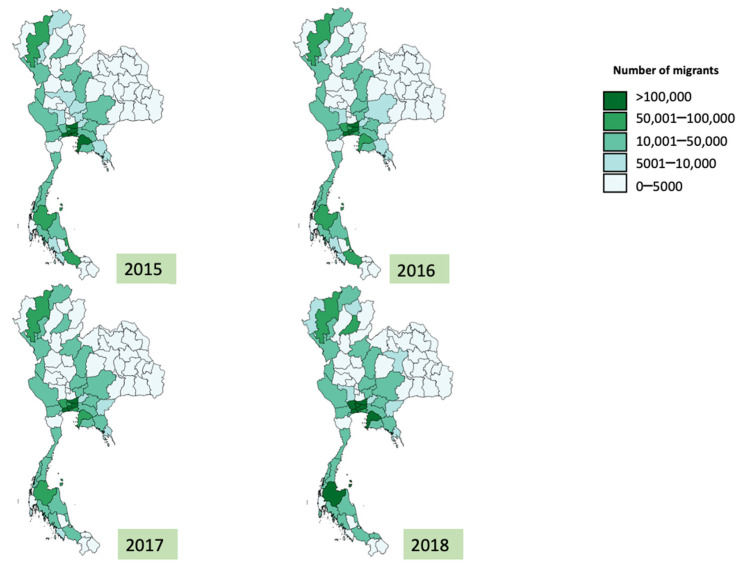
Geographical Difference of total migrants’ number, 2015–2018.

**Figure 3 ijerph-19-00064-f003:**
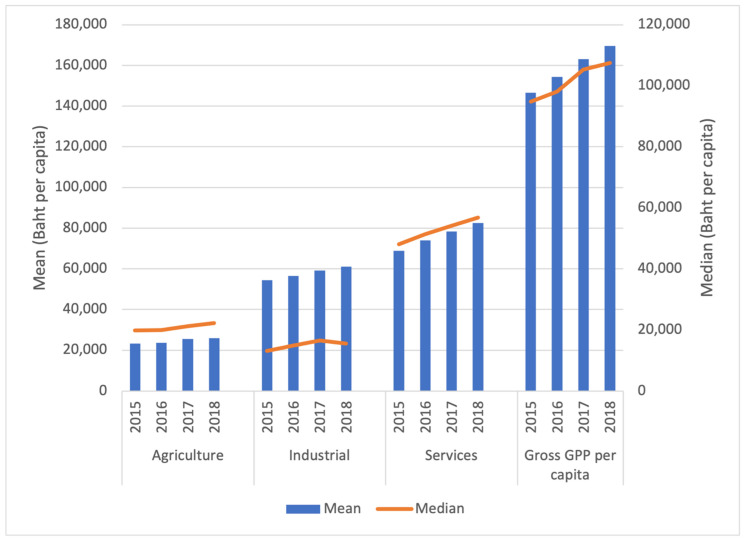
Mean and median of provincial economic variables in Thailand during 2015–2018. Note: $1 US Dollar = 33.89 Baht (as of 8 October 2021).

**Figure 4 ijerph-19-00064-f004:**
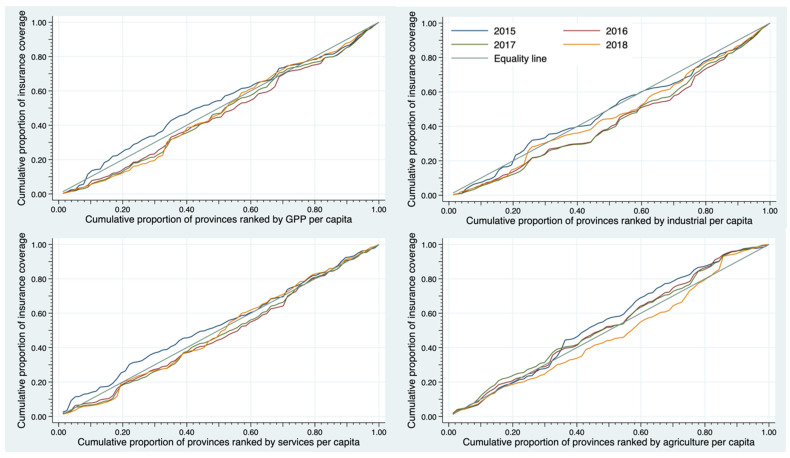
Concentration curve of SSS coverage among all migrants and GPP per capita and concentration curve of SSS coverage and the rank of provincial economic contribution by industrial sector, service sector, and agricultural sector, 2015 to 2018.

**Figure 5 ijerph-19-00064-f005:**
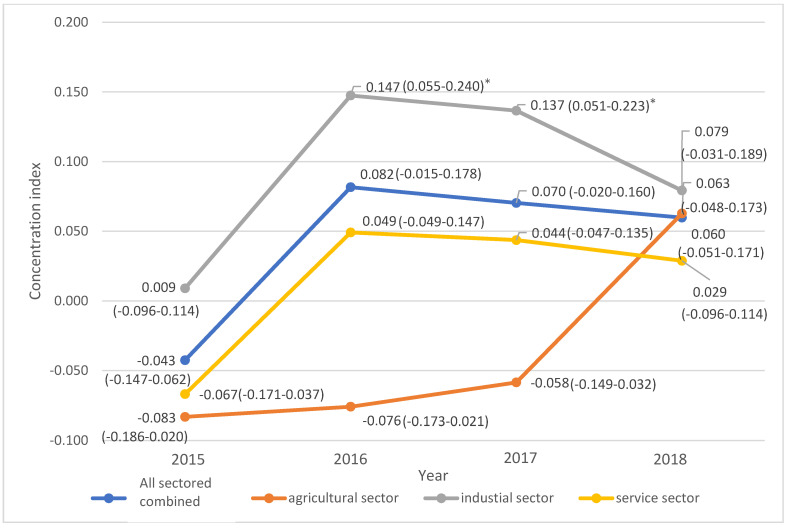
Showed the trend of concentration index in agricultural, industrial, and service sectors and total migrants by year 2015 to 2018. * Statistical Significance above 99% confidence level.

**Table 1 ijerph-19-00064-t001:** Number of all migrants, number of SSS migrants, and the coverage of SSS to all migrants across all provinces in Thailand during 2015–2018.

Year	All Migrants across Provinces		SSS Insurees across Provinces		Ratio of SSS Insurees to All Migrant across Provinces	
	Mean (SD)	Median (min–max)	Mean (SD)	Median (min–max)	Mean (SD)	Median (min–max)
2015	19,527 (36,462)	4210 (41–166,021)	6391 (16,614)	912 (7–100,663)	38.5% (31%)	31.9% (0.03–1.76)
2016	19,798 (41,012)	4056 (87–259,442)	6337 (16,345)	823 (10–105,600)	32.6% (22%)	29% (0.03–1.14)
2017	20,804 (43,122)	4598 (102–252,871)	6619 (17,101)	991 (9–113,602)	33.9% (24%)	28.2% (0.02–0.99)
2018	29,224 (65,787)	7114 (216–459,118)	14,785 (35,009)	2445 (50–236,207)	58.9% (52%)	43.4% (0.08–3.66)

**Table 2 ijerph-19-00064-t002:** Spearman’s correlation coefficients between SSS coverage among all migrants and economic level by all sectors, during 2015–2018.

Parameters	2015	2016	2017	2018
GPP per capita	−0.0165	0.21	0.225 ^a^	0.237 ^a^
Agriculture product per capita	−0.208	−0.226 ^a^	−0.197	0.017
Industrial product per capita	0.122	0.374 ^b^	0.411 ^b^	0.346 ^c^
Services product per capita	−0.068	0.125	0.157	0.173

^a^ *p* value < 0.05, ^b^ *p* value < 0.01, ^c^ *p* value < 0.001.

## Data Availability

The datasets used and/or analysed during the current study are available from the corresponding author on reasonable request.

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
