# Peer review of "Equity of Social Health Insurance Coverage for Migrants in Thailand: A Concentration Index Analysis"

_ijerph, 2021, doi:10.3390/ijerph19010064_

Round 1

Reviewer 1 Report

The manuscript is very concise. From the beginning to the end, the author consistently pursues the goal.

I think that the introduction should be improved and enriched with more references.

I do not understand why in the list of references there are no names of authors - there are only initials of names and surnames.

Several times the author uses "on the other hand" but there is no "on the one hand".

Reviewer 2 Report

Thanks for giving me this chance to review this article. The research title is interesting but the article needs beefing up, especially to fit into the international audience’s scope and interests. What can the international audience learn from this case study from Thailand? What’s the policy implication for other countries? Can you provide more international literature on this topic, to make the case from Thailand more comparable to other similar case studies?

Abstract: Please specify the background and importance of your research at the initial of the abstract (using one sentence). And, what is the full name of CIs in the abstract?

Introduction: there are two main sources of migrants: international or internal migration. Do you mean international migration? Migrant workers in Thai-34 land are mostly from Myanmar, Lao PDR, Cambodia, and Vietnam. Please specify which kind of migration it is.

Line 79: The unit of analysis was province—How many provinces (spatial units), and how many administrative ranks of them?

Line 128: why did the number of SSS insurees increase dramatically in most 128 provinces from 2015 to 2018 (Figure 1, especially from 2017 to 2018)? Please explain briefly using short sentences.

Table 1: does it fit into the table format (e.g. three-line tables of scientific papers)?

Line 156: please specify the exchange ratio of Baht to US$.

Line 169: why is there a positive relationship? Does it imply that the rapid industrialization or positive industrial policy can enhance the SSS coverage? What is the implication of it?

Figs 4-7: would you put them (four) into a combined figure—two above, and the other two below? I guess this would be more comparative. Can you also summarize more clearly the convergence/divergence of the main findings from the four figures?

Figure 8: what is the implication from the short findings of Figure 8? Which sector is more important, or which is more different…?

Discussion: I suggest putting the background information (such as policy change) to the introduction, research question, and data analysis, too. If you show the data first, and then introduce the policy, I am afraid that the article looks like a data report with a short policy analysis. But the academic interest would be the policy itself, its impact, its geographical difference, and its relations with different regional economies and sector situations. The policy implication for the broader other developing countries is as important as the data itself. Discussion can include the policy comparison (if any, such as the previous development stage, or other neighboring countries with similar development challenges in Southeastern Asia). Discussion can also tell more details about the different applications of the policy in different provinces that can lead to the different effects. I guess the different economic levels of provinces only tell a partial story of the regional difference in social health insurance coverage for migrants. Why is there a difference between different sectors? Would the government take different actions for different sectors, since that there are different trends of the concentration index in agricultural, industrial, and service sectors (Figure 8)?

Conclusions: It is a bit short, compared to other articles.

Reviewer 3 Report

The article presents an important research question related to the Social Security Scheme coverage equity among migrant workers in Thailand, sorted by businesses (agriculture, services and industrial sectors) and Gross Provincial Product (GPP) per capita.

The article is well written and contributes to social health insurance coverage equity for migrants in Thailand literature. However, I have a few suggestions for improvement:

Restructure the abstract. There are many repetitions of the period. I suggest that the period be synthesized only once (2015-2018).

In results highlight the practical, theoretical, and political implications of the study.

Round 2

Reviewer 2 Report

Agree for publication.